# Structure Regularization for Structured Prediction

**Xu Sun**[*†]
[*]MOE Key Laboratory of Computational Linguistics, Peking University
[†]School of Electronics Engineering and Computer Science, Peking University
xusun@pku.edu.cn

## Abstract

While there are many studies on weight regularization, the study on structure regularization is rare. Many existing systems on structured prediction focus on increasing the level of structural dependencies within the model. However, this trend could have been misdirected, because our study suggests that complex structures are actually harmful to generalization ability in structured prediction. To control structure-based overfitting, we propose a structure regularization framework via *structure decomposition*, which decomposes training samples into mini-samples with simpler structures, deriving a model with better generalization power. We show both theoretically and empirically that structure regularization can effectively control overfitting risk and lead to better accuracy. As a by-product, the proposed method can also substantially accelerate the training speed. The method and the theoretical results can apply to general graphical models with arbitrary structures. Experiments on well-known tasks demonstrate that our method can easily beat the benchmark systems on those highly-competitive tasks, achieving record-breaking accuracies yet with substantially faster training speed.

## 1 Introduction

Structured prediction models are popularly used to solve structure dependent problems in a wide variety of application domains including natural language processing, bioinformatics, speech recognition, and computer vision. Recently, many existing systems on structured prediction focus on increasing the level of structural dependencies within the model. We argue that this trend could have been misdirected, because our study suggests that complex structures are actually harmful to model accuracy. While it is obvious that intensive structural dependencies can effectively incorporate structural information, it is less obvious that intensive structural dependencies have a drawback of increasing the generalization risk, because more complex structures are easier to suffer from overfitting. Since this type of overfitting is caused by structure complexity, it can hardly be solved by ordinary regularization methods such as $L_2$ and $L_1$ regularization schemes, which is only for controlling weight complexity.

To deal with this problem, we propose a simple structure regularization solution based on *tag structure decomposition*. The proposed method decomposes each training sample into multiple mini-samples with simpler structures, deriving a model with better generalization power. The proposed method is easy to implement, and it has several interesting properties: (1) We show both theoretically and empirically that the proposed method can effectively reduce the overfitting risk on structured prediction. (2) The proposed method does not change the convexity of the objective function, such that a convex function penalized with a structure regularizer is still convex. (3) The proposed method has no conflict with the weight regularization. Thus we can apply structure regularization together with weight regularization. (4) The proposed method can accelerate the convergence rate in training.

The term *structural regularization* has been used in prior work for regularizing *structures of features*, including spectral regularization [1], regularizing feature structures for classifiers [20], and many

recent studies on structured sparsity in structured prediction scenarios [11, 8], via adopting mixed norm regularization [10], *Group Lasso* [22], and posterior regularization [5]. Compared with those prior work, we emphasize that our proposal on tag structure regularization is novel. This is because the term *structure* in all of the aforementioned work refers to *structures of feature space*, which is substantially different compared with our proposal on regularizing tag structures (interactions among tags).

Also, there are some other related studies. [17] described an interesting heuristic piecewise training method. [19] described a "lookahead" learning method. Our work differs from [17] and [19] mainly because our work is built on a regularization framework, with arguments and theoretical justifications on reducing generalization risk and improving convergence rate. Also, our method and the theoretical results can fit general graphical models with arbitrary structures, and the detailed algorithm is very different. On generalization risk analysis, related studies include [2, 12] on non-structured classification and [18, 7] on structured classification.

To the best of our knowledge, this is the first theoretical result on quantifying the relation between structure complexity and the generalization risk in structured prediction, and this is also the first proposal on structure regularization via regularizing tag-interactions. The contributions of this work[1] are two-fold:

- On the methodology side, we propose a structure regularization framework for structured prediction. We show both theoretically and empirically that the proposed method can effectively reduce the overfitting risk, and at the same time accelerate the convergence rate in training. Our method and the theoretical analysis do *not* make assumptions based on specific structures. In other words, the method and the theoretical results can apply to graphical models with arbitrary structures, including linear chains, trees, and general graphs.

- On the application side, for several important natural language processing tasks, our simple method can easily beat the benchmark systems on those highly-competitive tasks, achieving record-breaking accuracies as well as substantially faster training speed.

## 2   Structure Regularization

A graph of observations (even with arbitrary structures) can be indexed and be denoted by using an indexed sequence of observations $\boldsymbol{O} = \{o_1, \ldots, o_n\}$. We use the term *sample* to denote $\boldsymbol{O} = \{o_1, \ldots, o_n\}$. For example, in natural language processing, a sample may correspond to a sentence of $n$ words with dependencies of tree structures (e.g., in syntactic parsing). For simplicity in analysis, we assume all samples have $n$ observations (thus $n$ tags). In a typical setting of structured prediction, all the $n$ tags have inter-dependencies via connecting each Markov dependency between neighboring tags. Thus, we call $n$ as *tag structure complexity* or simply *structure complexity* below.

A sample is converted to an indexed sequence of feature vectors $\boldsymbol{x} = \{\boldsymbol{x}_{(1)}, \ldots, \boldsymbol{x}_{(n)}\}$, where $\boldsymbol{x}_{(k)} \in \mathcal{X}$ is of the dimension $d$ and corresponds to the local features extracted from the position/index $k$. We can use an $n \times d$ matrix to represent $\boldsymbol{x} \in \mathcal{X}^n$. Let $\mathcal{Z} = (\mathcal{X}^n, \mathcal{Y}^n)$ and let $\boldsymbol{z} = (\boldsymbol{x}, \boldsymbol{y}) \in \mathcal{Z}$ denote a sample in the training data. Suppose a training set is $S = \{\boldsymbol{z}_1 = (\boldsymbol{x}_1, \boldsymbol{y}_1), \ldots, \boldsymbol{z}_m = (\boldsymbol{x}_m, \boldsymbol{y}_m)\}$, with size $m$, and the samples are drawn i.i.d. from a distribution $D$ which is unknown. A learning algorithm is a function $G : \mathcal{Z}^m \mapsto \mathcal{F}$ with the function space $\mathcal{F} \subset \{\mathcal{X}^n \mapsto \mathcal{Y}^n\}$, i.e., $G$ maps a training set $S$ to a function $G_S : \mathcal{X}^n \mapsto \mathcal{Y}^n$. We suppose $G$ is symmetric with respect to $S$, so that $G$ is independent on the order of $S$.

Structural dependencies among tags are the major difference between structured prediction and non-structured classification. For the latter case, a local classification of $g$ based on a position $k$ can be expressed as $g(\boldsymbol{x}_{(k-a)}, \ldots, \boldsymbol{x}_{(k+a)})$, where the term $\{\boldsymbol{x}_{(k-a)}, \ldots, \boldsymbol{x}_{(k+a)}\}$ represents a local window. However, for structured prediction, a local classification on a position depends on the whole input $\boldsymbol{x} = \{\boldsymbol{x}_{(1)}, \ldots, \boldsymbol{x}_{(n)}\}$ rather than a local window, due to the nature of structural dependencies among tags (e.g., graphical models like CRFs). Thus, in structured prediction a local classification on $k$ should be denoted as $g(\boldsymbol{x}_{(1)}, \ldots, \boldsymbol{x}_{(n)}, k)$. To simplify the notation, we define

$$g(\boldsymbol{x}, k) \triangleq g(\boldsymbol{x}_{(1)}, \ldots, \boldsymbol{x}_{(n)}, k)$$

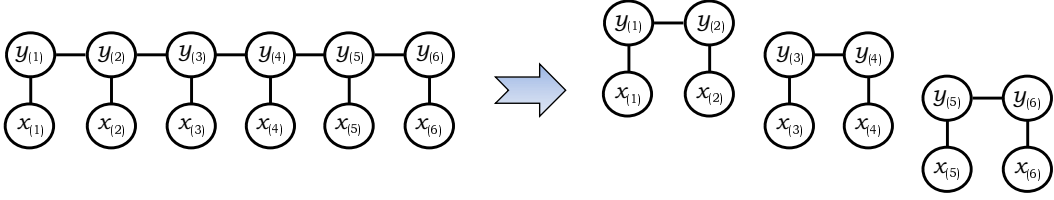

Figure 1: An illustration of structure regularization in simple linear chain case, which decompose a training sample $\boldsymbol{z}$ with structure complexity 6 into three mini-samples with structure complexity 2. Structure regularization can apply to more general graphs with arbitrary dependencies.

We define *point-wise cost function* $c : \mathcal{Y} \times \mathcal{Y} \mapsto \mathbb{R}^+$ as $c[G_S(\boldsymbol{x}, k), \boldsymbol{y}_{(k)}]$, which measures the cost on a position $k$ by comparing $G_S(\boldsymbol{x}, k)$ and the gold-standard tag $\boldsymbol{y}_{(k)}$, and we introduce the point-wise loss as

$$\ell(G_S, \boldsymbol{z}, k) \triangleq c[G_S(\boldsymbol{x}, k), \boldsymbol{y}_{(k)}]$$

Then, we define *sample-wise cost function* $C : \mathcal{Y}^n \times \mathcal{Y}^n \mapsto \mathbb{R}^+$, which is the cost function with respect to a whole sample, and we introduce the sample-wise loss as

$$\mathcal{L}(G_S, \boldsymbol{z}) \triangleq C[G_S(\boldsymbol{x}), \boldsymbol{y}] = \sum_{k=1}^{n} \ell(G_S, \boldsymbol{z}, k) = \sum_{k=1}^{n} c[G_S(\boldsymbol{x}, k), \boldsymbol{y}_{(k)}]$$

Given $G$ and a training set $S$, what we are most interested in is the *generalization risk* in structured prediction (i.e., expected average loss) [18, 7]:

$$R(G_S) = \mathbb{E}_{\boldsymbol{z}} \left[ \frac{\mathcal{L}(G_S, \boldsymbol{z})}{n} \right]$$

Since the distribution $D$ is unknown, we have to estimate $R(G_S)$ by using the *empirical risk*:

$$R_e(G_S) = \frac{1}{mn} \sum_{i=1}^{m} \mathcal{L}(G_S, \boldsymbol{z}_i) = \frac{1}{mn} \sum_{i=1}^{m} \sum_{k=1}^{n} \ell(G_S, \boldsymbol{z}_i, k)$$

To state our theoretical results, we must describe several quantities and assumptions following prior work [2, 12]. We assume a simple real-valued structured prediction scheme such that the class predicted on position $k$ of $\boldsymbol{x}$ is the sign of $G_S(\boldsymbol{x}, k) \in \mathcal{D}$.[2] Also, we assume the point-wise cost function $c_\tau$ is convex and $\tau$-*smooth* such that $\forall y_1, y_2 \in \mathcal{D}, \forall y^* \in \mathcal{Y}$

$$|c_\tau(y_1, y^*) - c_\tau(y_2, y^*)| \le \tau|y_1 - y_2| \tag{1}$$

Also, we use a value $\rho$ to quantify the bound of $|G_S(\boldsymbol{x}, k) - G_{S \setminus i}(\boldsymbol{x}, k)|$ while changing a single sample (with size $n' \le n$) in the training set with respect to the structured input $\boldsymbol{x}$. This $\rho$-*admissible* assumption can be formulated as $\forall k$,

$$|G_S(\boldsymbol{x}, k) - G_{S \setminus i}(\boldsymbol{x}, k)| \le \rho ||G_S - G_{S \setminus i}||_2 \cdot ||\boldsymbol{x}||_2 \tag{2}$$

where $\rho \in \mathbb{R}^+$ is a value related to the design of algorithm $G$.

## 2.1 Structure Regularization

Most existing regularization techniques are for regularizing model weights/parameters (e.g., a representative regularizer is the Gaussian regularizer or so called $L_2$ regularizer), and we call such regularization techniques as *weight regularization*.

**Definition 1 (Weight regularization)** *Let $N_\lambda : \mathcal{F} \mapsto \mathbb{R}^+$ be a weight regularization function on $\mathcal{F}$ with regularization strength $\lambda$, the structured classification based objective function with general weight regularization is as follows:*

$$R_\lambda(G_S) \triangleq R_e(G_S) + N_\lambda(G_S) \tag{3}$$

**Algorithm 1** Training with structure regularization
---
1: **Input**: model weights $\boldsymbol{w}$, training set $S$, structure regularization strength $\alpha$
2: **repeat**
3:     $S' \leftarrow \emptyset$
4:     **for** $i = 1 \rightarrow m$ **do**
5:         Randomly decompose $\boldsymbol{z}_i \in S$ into mini-samples $N_\alpha(\boldsymbol{z}_i) = \{\boldsymbol{z}_{(i,1)}, \ldots, \boldsymbol{z}_{(i,\alpha)}\}$
6:         $S' \leftarrow S' \cup N_\alpha(\boldsymbol{z}_i)$
7:     **end for**
8:     **for** $i = 1 \rightarrow |S'|$ **do**
9:         Sample $\boldsymbol{z}'$ uniformly at random from $S'$, with gradient $\nabla g_{\boldsymbol{z}'}(\boldsymbol{w})$
10:         $\boldsymbol{w} \leftarrow \boldsymbol{w} - \eta \nabla g_{\boldsymbol{z}'}(\boldsymbol{w})$
11:     **end for**
12: **until** Convergence
13: **return** $\boldsymbol{w}$
---

While weight regularization is normalizing model weights, the proposed structure regularization method is normalizing the structural complexity of the training samples. As illustrated in Figure 1, our proposal is based on *tag structure decomposition*, which can be formally defined as follows:

**Definition 2 (Structure regularization)** *Let $N_\alpha : \mathcal{F} \mapsto \mathcal{F}$ be a structure regularization function on $\mathcal{F}$ with regularization strength $\alpha$ with $1 \leq \alpha \leq n$, the structured classification based objective function with structure regularization is as follows[3]:*

$$R_\alpha(G_S) \triangleq R_e[G_{N_\alpha(S)}] = \frac{1}{mn} \sum_{i=1}^{m} \sum_{j=1}^{\alpha} \mathcal{L}[G_{S'}, \boldsymbol{z}_{(i,j)}] = \frac{1}{mn} \sum_{i=1}^{m} \sum_{j=1}^{\alpha} \sum_{k=1}^{n/\alpha} \ell[G_{S'}, \boldsymbol{z}_{(i,j)}, k] \quad (4)$$

*where $N_\alpha(\boldsymbol{z}_i)$ randomly splits $\boldsymbol{z}_i$ into $\alpha$ mini-samples $\{\boldsymbol{z}_{(i,1)}, \ldots, \boldsymbol{z}_{(i,\alpha)}\}$, so that the mini-samples have a distribution on their sizes (structure complexities) with the expected value $n' = n/\alpha$. Thus, we get*

$$S' = \{\underbrace{\boldsymbol{z}_{(1,1)}, z_{(1,2)}, \ldots, \boldsymbol{z}_{(1,\alpha)}}_{\alpha}), \ldots, \underbrace{\boldsymbol{z}_{(m,1)}, \boldsymbol{z}_{(m,2)}, \ldots, \boldsymbol{z}_{(m,\alpha)}}_{\alpha}\} \quad (5)$$

*with $m\alpha$ mini-samples with expected structure complexity $n/\alpha$. We can denote $S'$ more compactly as $S' = \{\boldsymbol{z}'_1, \boldsymbol{z}'_2, \ldots, \boldsymbol{z}'_{m\alpha}\}$ and $R_\alpha(G_S)$ can be simplified as*

$$R_\alpha(G_S) \triangleq \frac{1}{mn} \sum_{i=1}^{m\alpha} \mathcal{L}(G_{S'}, \boldsymbol{z}'_i) = \frac{1}{mn} \sum_{i=1}^{m\alpha} \sum_{k=1}^{n/\alpha} \ell[G_{S'}, \boldsymbol{z}'_i, k] \quad (6)$$

When the structure regularization strength $\alpha = 1$, we have $S' = S$ and $R_\alpha = R_e$. The structure regularization algorithm (with the stochastic gradient descent setting) is summarized in Algorithm 1. Recall that $\boldsymbol{x} = \{\boldsymbol{x}_{(1)}, \ldots, \boldsymbol{x}_{(n)}\}$ represents feature vectors. Thus, it should be emphasized that the decomposition of $\boldsymbol{x}$ is the decomposition of the feature vectors, not the original observations. Actually the decomposition of the feature vectors is more convenient and has no information loss — decomposing observations needs to regenerate features and may lose some features.

The structure regularization has no conflict with the weight regularization, and the structure regularization can be applied together with the weight regularization.

**Definition 3 (Structure & weight regularization)** *By combining structure regularization in Definition 2 and weight regularization in Definition 1, the structured classification based objective function is as follows:*

$$R_{\alpha,\lambda}(G_S) \triangleq R_\alpha(G_S) + N_\lambda(G_S) \quad (7)$$

*When $\alpha = 1$, we have $R_{\alpha,\lambda} = R_e(G_S) + N_\lambda(G_S) = R_\lambda$.*

Like existing weight regularization methods, currently our structure regularization is only for the training stage. Currently we do not use structure regularization in the test stage.

## 2.2 Reduction of Generalization Risk

In contrast to the simplicity of the algorithm, the theoretical analysis is quite technical. In this paper we only describe the major theoretical result. Detailed analysis and proofs are given in the full version of this work [14].

**Theorem 4 (Generalization vs. structure regularization)** *Let the structured prediction objective function of $G$ be penalized by structure regularization with factor $\alpha \in [1, n]$ and $L_2$ weight regularization with factor $\lambda$, and the penalized function has a minimizer $f$:*

$$f = \underset{g \in \mathcal{F}}{\operatorname{argmin}} R_{\alpha, \lambda}(g) = \underset{g \in \mathcal{F}}{\operatorname{argmin}} \Big( \frac{1}{mn} \sum_{j=1}^{m\alpha} \mathcal{L}_\tau(g, \boldsymbol{z}'_j) + \frac{\lambda}{2} ||g||_2^2 \Big) \tag{8}$$

*Assume the point-wise loss $\ell_\tau$ is convex and differentiable, and is bounded by $\ell_\tau(f, \boldsymbol{z}, k) \leq \gamma$. Assume $f(\boldsymbol{x}, k)$ is $\rho$-admissible. Let a local feature value be bounded by $v$ such that $\boldsymbol{x}_{(k,q)} \leq v$ for $q \in \{1, \ldots, d\}$. Then, for any $\delta \in (0, 1)$, with probability at least $1 - \delta$ over the random draw of the training set $S$, the generalization risk $R(f)$ is bounded by*

$$R(f) \leq R_e(f) + \frac{2d\tau^2 \rho^2 v^2 n^2}{m\lambda\alpha} + \Big( \frac{(4m-2)d\tau^2 \rho^2 v^2 n^2}{m\lambda\alpha^2} + \gamma \Big) \sqrt{\frac{\alpha \ln \delta^{-1}}{2m}} \tag{9}$$

*Since $\tau, \rho$, and $v$ are typically small compared with other variables, especially $m$, (9) can be approximated as follows by ignoring small terms:*

$$R(f) \leq R_e(f) + O\Big( \frac{dn^2 \sqrt{\ln \delta^{-1}}}{\lambda \alpha^{1.5} \sqrt{m}} \Big) \tag{10}$$

The proof is given in the full version of this work [14]. We call the term $O\Big( \frac{dn^2 \sqrt{\ln \delta^{-1}}}{\lambda \alpha^{1.5} \sqrt{m}} \Big)$ in (10) as "overfit-bound", and reducing the overfit-bound is crucial for reducing the generalization risk bound. First, (10) suggests that structure complexity $n$ can increase the overfit-bound on a magnitude of $O(n^2)$, and applying weight regularization can reduce the overfit-bound by $O(\lambda)$. Importantly, applying structure regularization further (over weight regularization) can additionally reduce the overfit-bound by a magnitude of $O(\alpha^{1.5})$. Since many applications in practice are based on sparse features, using a sparse feature assumption can further improve the generalization bound. The improved generalization bounds are given in the full version of this work [14].

## 2.3 Accelerating Convergence Rates in Training

We also analyze the impact on the convergence rate of online learning by applying structure regularization. Following prior work [9], our analysis is based on the stochastic gradient descent (SGD) with fixed learning rate. Let $g(\boldsymbol{w})$ be the structured prediction objective function and $\boldsymbol{w} \in \mathcal{W}$ is the weight vector. Recall that the SGD update with fixed learning rate $\eta$ has a form like this:

$$\boldsymbol{w}_{t+1} \leftarrow \boldsymbol{w}_t - \eta \nabla g_{\boldsymbol{z}_t}(\boldsymbol{w}_t) \tag{11}$$

where $g_{\boldsymbol{z}}(\boldsymbol{w}_t)$ is the stochastic estimation of the objective function based on $\boldsymbol{z}$ which is randomly drawn from $S$. To state our convergence rate analysis results, we need several assumptions following (Nemirovski et al. 2009). We assume $g$ is strongly convex with modulus $c$, that is, $\forall \boldsymbol{w}, \boldsymbol{w}' \in \mathcal{W}$,

$$g(\boldsymbol{w}') \geq g(\boldsymbol{w}) + (\boldsymbol{w}' - \boldsymbol{w})^T \nabla g(\boldsymbol{w}) + \frac{c}{2} ||\boldsymbol{w}' - \boldsymbol{w}||^2 \tag{12}$$

When $g$ is strongly convex, there is a global optimum/minimizer $\boldsymbol{w}^*$. We also assume Lipschitz continuous differentiability of $g$ with the constant $q$, that is, $\forall \boldsymbol{w}, \boldsymbol{w}' \in \mathcal{W}$,

$$||\nabla g(\boldsymbol{w}') - \nabla g(\boldsymbol{w})|| \leq q||\boldsymbol{w}' - \boldsymbol{w}|| \tag{13}$$

It is also reasonable to assume that the norm of $\nabla g_{\boldsymbol{z}}(\boldsymbol{w})$ has almost surely positive correlation with the structure complexity of $\boldsymbol{z}$,[4] which can be quantified by a bound $\kappa \in \mathbb{R}^+$:

$$||\nabla g_{\boldsymbol{z}}(\boldsymbol{w})||_2 \leq \kappa |\boldsymbol{z}| \quad \text{almost surely for} \quad \forall \boldsymbol{w} \in \mathcal{W} \tag{14}$$

where $|\boldsymbol{z}|$ denotes the structure complexity of $\boldsymbol{z}$. Moreover, it is reasonable to assume

$$\eta c < 1 \tag{15}$$

because even the ordinary gradient descent methods will diverge if $\eta c > 1$. Then, we show that structure regularization can quadratically accelerate the SGD rates of convergence:

**Proposition 5 (Convergence rates vs. structure regularization)** *With the aforementioned assumptions, let the SGD training have a learning rate defined as $\eta = \frac{c\epsilon\beta\alpha^2}{q\kappa^2 n^2}$, where $\epsilon > 0$ is a convergence tolerance value and $\beta \in (0, 1]$. Let t be a integer satisfying*

$$t \geq \frac{q\kappa^2 n^2 \log\left(qa_0/\epsilon\right)}{\epsilon\beta c^2\alpha^2} \tag{16}$$

*where $n$ and $\alpha \in [1, n]$ is like before, and $a_0$ is the initial distance which depends on the initialization of the weights $\boldsymbol{w}_0$ and the minimizer $\boldsymbol{w}^*$, i.e., $a_0 = ||\boldsymbol{w}_0 - \boldsymbol{w}^*||^2$. Then, after t updates of $\boldsymbol{w}$ it converges to $\mathbb{E}[g(\boldsymbol{w}_t) - g(\boldsymbol{w}^*)] \leq \epsilon$.*

The proof is given in the full version of this work [14]. As we can see, using structure regularization with the strength $\alpha$ can quadratically accelerate the convergence rate with a factor of $\alpha^2$.

## 3    Experiments

**Diversified Tasks.** The natural language processing tasks include (1) part-of-speech tagging, (2) biomedical named entity recognition, and (3) Chinese word segmentation. The signal processing task is (4) sensor-based human activity recognition. The tasks (1) to (3) use boolean features and the task (4) adopts real-valued features. From tasks (1) to (4), the averaged structure complexity (number of observations) $n$ is very different, with $n = 23.9, 26.5, 46.6, 67.9$, respectively. The dimension of tags $|\mathcal{Y}|$ is also diversified among tasks, with $|\mathcal{Y}|$ ranging from 5 to 45.

**Part-of-Speech Tagging (POS-Tagging).** Part-of-Speech (POS) tagging is an important and highly competitive task. We use the standard benchmark dataset in prior work [3], with 38,219 training samples and 5,462 test samples. Following prior work [19], we use features based on words and lexical patterns, with 393,741 raw features[5]. The evaluation metric is per-word accuracy.

**Biomedical Named Entity Recognition (Bio-NER).** This task is from the *BioNLP-2004 shared task* [19]. There are 17,484 training samples and 3,856 test samples. Following prior work [19], we use word pattern features and POS features, with 403,192 raw features in total. The evaluation metric is balanced F-score.

**Word Segmentation (Word-Seg).** We use the MSR data provided by *SIGHAN-2004 contest* [4]. There are 86,918 training samples and 3,985 test samples. The features are similar to [16], with 1,985,720 raw features in total. The evaluation metric is balanced F-score.

**Sensor-based Human Activity Recognition (Act-Recog).** This is a task based on real-valued sensor signals, with the data extracted from the Bao04 activity recognition dataset [15]. The features are similar to [15], with 1,228 raw features in total. There are 16,000 training samples and 4,000 test samples. The evaluation metric is accuracy.

We choose the CRFs [6] and structured perceptrons (Perc) [3], which are arguably the most popular probabilistic and non-probabilistic structured prediction models, respectively. The CRFs are trained using the SGD algorithm,[6] and the baseline method is the traditional weight regularization scheme (*WeightReg*), which adopts the most representative $L_2$ weight regularization, i.e., a Gaussian prior.[7] For the structured perceptrons, the baseline *WeightAvg* is the popular implicit regularization technique based on parameter averaging, i.e., *averaged perceptron* [3].

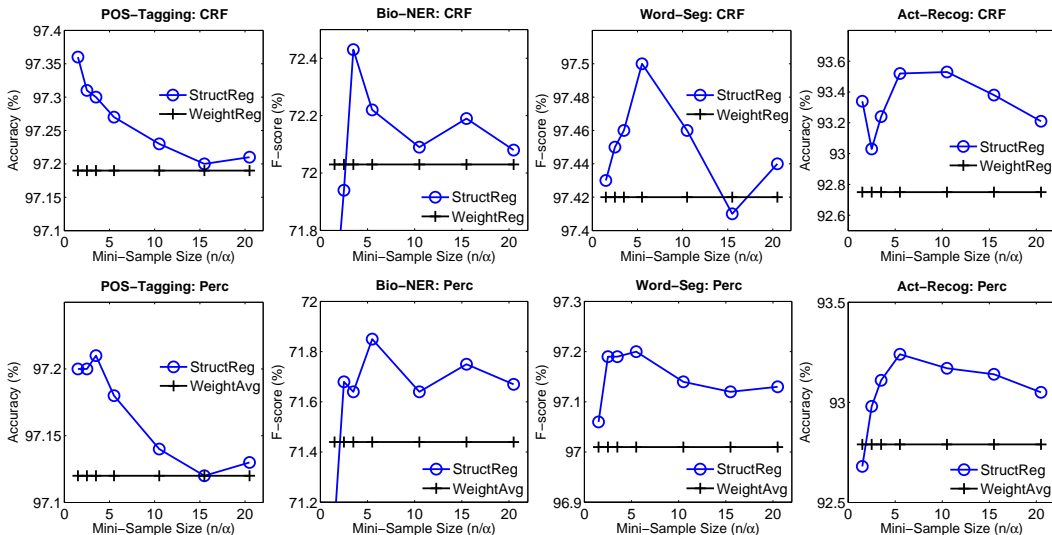

Figure 2: On the four tasks, comparing the structure regularization method (*StructReg*) with existing regularization methods in terms of accuracy/F-score. Row-1 shows the results on CRFs and Row-2 shows the results on structured perceptrons.

Table 1: Comparing our results with the benchmark systems on corresponding tasks.

|                   | POS-Tagging (Acc%) | Bio-NER (F1%)    | Word-Seg (F1%)  |
|-------------------|--------------------|------------------|-----------------|
| Benchmark system  | 97.33 (see [13])   | 72.28 (see [19]) | 97.19 (see [4]) |
| Our results       | **97.36**          | **72.43**        | **97.50**       |

The rich edge features [16] are employed for all methods. All methods are based on the 1st-order Markov dependency. For *WeightReg*, the $L_2$ regularization strengths (i.e., $\lambda/2$ in Eq.(8)) are tuned among values $0.1, 0.5, 1, 2, 5$, and are determined on the development data (POS-Tagging) or simply via 4-fold cross validation on the training set (Bio-NER, Word-Seg, and Act-Recog). With this automatic tuning for *WeightReg*, we set $2, 5, 1$ and $5$ for POS-Tagging, Bio-NER, Word-Seg, and Act-Recog tasks, respectively.

## 3.1 Experimental Results

The experimental results in terms of accuracy/F-score are shown in Figure 2. For the CRF model, the training is convergent, and the results on the convergence state (decided by relative objective change with the threshold value of $0.0001$) are shown. For the structured perceptron model, the training is typically not convergent, and the results on the 10'th iteration are shown. For stability of the curves, the results of the structured perceptrons are averaged over 10 repeated runs.

Since different samples have different size $n$ in practice, we set $\alpha$ being a function of $n$, so that the generated mini-samples are with *fixed* size $n'$ with $n' = n/\alpha$. Actually, $n'$ is a probabilistic distribution because we adopt randomized decomposition. For example, if $n' = 5.5$, it means the mini-samples are a mixture of the ones with the size 5 and the ones with the size 6, and the mean of the size distribution is 5.5. In the figure, the curves are based on $n' = 1.5, 2.5, 3.5, 5.5, 10.5, 15.5, 20.5$.

As we can see, the results are quite consistent. It demonstrates that structure regularization leads to higher accuracies/F-scores compared with the existing baselines. We also conduct significance tests based on t-test. Since the t-test for F-score based tasks (Bio-NER and Word-Seg) may be unreliable[8], we only perform t-test for the accuracy-based tasks, i.e., POS-Tagging and Act-Recog. For POS-Tagging, the significance test suggests that the superiority of *StructReg* over *WeightReg* is very statistically significant, with $p < 0.01$. For Act-Recog, the significance tests suggest that both the *StructReg vs. WeightReg* difference and the *StructReg vs. WeightAvg* difference are extremely statis-

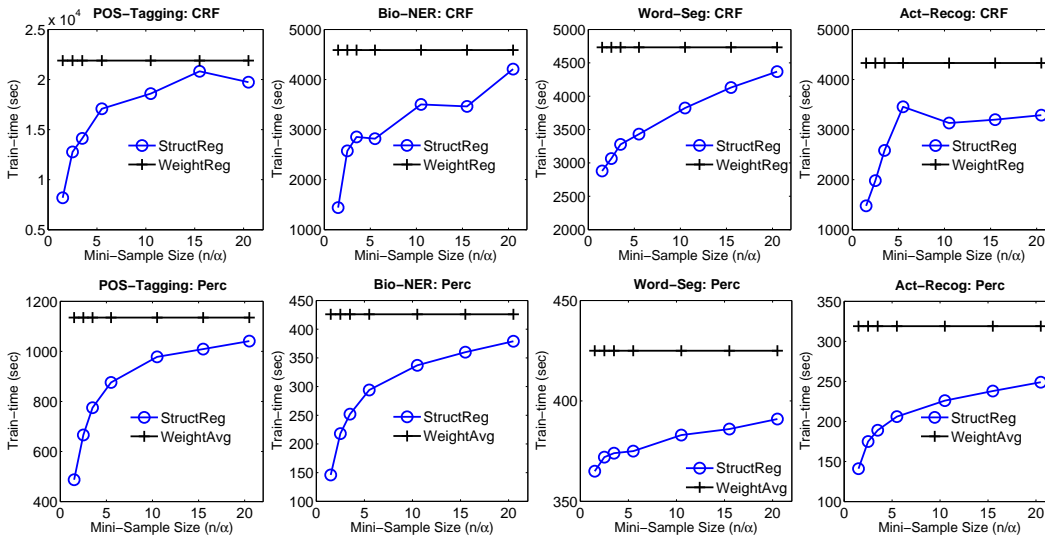

Figure 3: On the four tasks, comparing the structure regularization method (*StructReg*) with existing regularization methods in terms of wall-clock training time.

tically significant, with $p < 0.0001$ in both cases. The experimental results support our theoretical analysis that structure regularization can further reduce the generalization risk over existing weight regularization techniques.

Our method outperforms the benchmark systems on the three important natural language processing tasks. The POS-Tagging task is a highly competitive task, with many methods proposed, and the best report (without using extra resources) until now is achieved by using a bidirectional learning model in [13],[9] with the accuracy 97.33%. Our simple method achieves better accuracy compared with all of those state-of-the-art systems. Furthermore, our method achieves as good scores as the benchmark systems on the Bio-NER and Word-Seg tasks. On the Bio-NER task, [19] achieves 72.28% based on lookahead learning and [21] achieves 72.65% based on reranking. On the Word-Seg task, [4] achieves 97.19% based on maximum entropy classification and our recent work [16] achieves 97.5% based on feature-frequency-adaptive online learning. The comparisons are summarized in Table 1.

Figure 3 shows experimental comparisons in terms of wall-clock training time. As we can see, the proposed method can substantially improve the training speed. The speedup is not only from the faster convergence rates, but also from the faster processing time on the structures, because it is more efficient to process the decomposed samples with simple structures.

## 4   Conclusions

We proposed a structure regularization framework, which decomposes training samples into mini-samples with simpler structures, deriving a trained model with regularized structural complexity. Our theoretical analysis showed that this method can effectively reduce the generalization risk, and can also accelerate the convergence speed in training. The proposed method does not change the convexity of the objective function, and can be used together with any existing weight regularization methods. Note that, the proposed method and the theoretical results can fit general structures including linear chains, trees, and graphs. Experimental results demonstrated that our method achieved better results than state-of-the-art systems on several highly-competitive tasks, and at the same time with substantially faster training speed.

**Acknowledgments**. This work was supported in part by NSFC (No.61300063).

## Footnotes

[1]See the code at `http://klcl.pku.edu.cn/member/sunxu/code.htm`

[2]In practice, many popular structured prediction models have a convex and real-valued cost function (e.g., CRFs).

[3]The notation $N$ is overloaded here. For clarity throughout, $N$ with subscript $\lambda$ refers to weight regularization function, and $N$ with subscript $\alpha$ refers to structure regularization function.

[4]Many structured prediction systems (e.g., CRFs) satisfy this assumption that the gradient based on a larger sample (i.e., $n$ is large) is expected to have a larger norm.

[5]Raw features are those observation features based only on $\boldsymbol{x}$, i.e., no combination with tag information.

[6]In theoretical analysis, following prior work we adopt the SGD with fixed learning rate, as described in Section 2.3. However, since the SGD with decaying learning rate is more commonly used in practice, in experiments we use the SGD with decaying learning rate.

[7]We also tested on sparsity emphasized regularization methods, including $L_1$ regularization and *Group Lasso* regularization [8]. However, we find that in most cases those sparsity emphasized regularization methods have lower accuracy than the $L_2$ regularization.

[8]Indeed we can convert F-scores to accuracy scores for t-test, but in many cases this conversion is unreliable. For example, very different F-scores may correspond to similar accuracy scores.

[9]See a collection of the systems at `http://aclweb.org/aclwiki/index.php?title=POS_Tagging_(State_of_the_art)`

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
