[Reviews · NeurIPS 2014]

Submitted by Assigned_Reviewer_1

This paper proposes a new regularization method for structured prediction. The idea is relatively straightforward: a linear chain model is segmented into smaller subchains, each of which is added as an independent training example.

Theorems are provided (with proofs in the supplement) showing how this regularization can reduce generalization risk and accelerate convergence rates. Empirical comparisons with state of the art approaches suggest that the resulting method is both faster and more accurate. The accuracy improvements are small, but these are all well-studied tasks where small improvements can have impact.

My primary concern is the assumption of a linear chain. Is this a limitation of the method? If not, the notation should be generalized to accept graphical models of arbitrary structure. Otherwise, the article should be written assuming a linear chain. There are obvious additional difficulties when considering more complex models --- even if the scope of this paper is limited to linear chains, I think this is still a sufficient contribution.

A secondary concern is that the paper should be better placed in the context of related work. How does this approach relate to other work that approximates structure at training time? E.g., piecewise training (Sutton & McCallum, ICML 2007) or Wainright's "Estimating the “Wrong” Graphical Model" paper (JMLR 2006). In general, please better situate this submission in the context of related work.

Finally, the paper could use a more thorough edit to remove some distracting errors and add additional clarifications. Some suggestions:

- Proposition 3: it is not clear why the learning rate is defined as such. Please motivate and clarify.

- Is Figure 2 accuracy or F1? The text seems to confuse these terms.

- I assume only the best performing StructReg results are used for computing significance? Surely it is not significant for all values of alpha.

058: confliction -> conflict
078: two-folds -> two-fold
098: local window of $o_k$: "Window" typically denotes features of adjacent nodes in a chain, but this notation suggests the features are of observation $o_k$. Please clarify.
138: draw -> drawn
164: Please define $N_\lambda$ in Eq. 1
193: I believe $g$ in Eq. 5 should be G, to be consistent with Eq. 4.
201: focus -> focuses
220: xvalue -> value (?)
229: value is bounded -> value be bounded
236: simplified as -> approximated by (?) Since you're ignoring small terms.

Summary: This paper proposes a new regularization method for structured prediction, providing theoretical and empirical evidence of its efficacy. While the approach is straightforward, it appears to perform quite well, and has an interesting theoretical foundation; I feel this paper is a valuable contribution to structured classification problems.

Submitted by Assigned_Reviewer_3

The paper investigates decomposing the structured output object into smaller objects in Structured Output prediction setting. Theoretical analysis is provided to show that the decomposition approach has better generalization then regular SO models. In case of CRF, convergence rates are shown to be faster for optimization with SGD. Experiments on sequence models with 4 different applications show superior/competitive (3/1) performance to the state-of-the art systems in the literature.

I find the theoretical analysis of the the decompositional approach refreshing, since there has been various proposal of this approach algorithmically without any theoretical analysis. One problem that I see is that the provided bounds are much looser than the existing analysis of SO prediction models for hinge-loss,
[Taskar et al, Max-Margin Markov Networks, NIPS 2003,
McAllister, Generalization bounds and consistency for Structured Labeling, in Predicting Structured Data, 2006]
where the generalization bound is log(l) opposed to l4 in this paper with l being the size of the structured object. Given that, it is not clear to me how the analysis of the decompositional approach would translate in the more informative analysis.

In the experimental side, the results look very impressive. In particular for segmentation tasks (NER and Word Segmentation), a consistent labeling of the segmentation is fairly important [in BIO terms, I(In) label does not mean anything if there is no B(Begin) term]. The figures seem to suggest that in Chinese word segmentation for CRFs, simply looking at the bigram labels yields the best performance (46.6/20) and similarly for NER (26.5/~10). [Can the authors pls comment whether this is a correct reading of the comparisons?]. I am assuming here, that during testing the same decomposition algorithm is applied. I have also notice that comparing Table 1, the performance of the model on the (test?) data, is very much aligned with the best values of the hyperparameter in Figure 2, which is stated to be set on development data. It would be great if the authors could provide some discussions on these two aspect.

The decomposition of the model is done randomly for each instance (Algorithm 1). Such decomposition naturally translates sequence models. Sequence modeling is still the most representative SO prediction problem. However, it would be great to provide the reader (some possibly informal discussion) what are the consequence of the random decomposition approach for models with clique size larger than 2.
Summary: A decompositional approach to SO prediction supported with (rather uninformative) theoretical analysis and strong empirical analysis.

Submitted by Assigned_Reviewer_41

Summary: The authors propose “tag structure regularization”, a novel regularization scheme whereby at training time, a structured prediction model is penalized based on losses on sub-structures of the training example. The key is that the model is forced to make predictions for each sub-structure independently, so it cannot rely on long-reaching information from other parts of the graph. They provide analysis based on stability arguments that the regularization strength decreases bounds on generalization error and increases the convergence rate of SGD. On several sequential prediction tasks they show that varying the regularization strength can produce state-of-the-art accuracies on several tasks and at the same time speed up training.

Major comments: (+ Pros, - Cons)

+ On the one hand, the central idea in this paper is very interesting. The authors argue that by breaking up training samples into sub-samples during learning, we can increase the generalization of structured models. Intuitively, this makes a certain kind of sense: the structured model is regularized to rely more on local information than incoming messages from the rest of the graph. This might prevent errors from propagating, and results in a “simpler” model. Another way to think about it is that if a model can make accurate predictions without relying on passing messages, the “tag structure regularization” will choose that simpler model, while more standard approaches do not have such preference.

- On the other hand, the actual theoretical analysis in this paper seems like it isn’t making the correct assumptions or taking the right approach to analysis. Yes, increasing alpha does reduce the bound -- but at the tightest setting alpha = n, the bound is still far looser than other structured prediction generalization bounds. E.g. the original MMMN paper (Taskar et al.) had a logarithmic complexity in both the multi class label size and the number of variables (l in that paper, n here). More recently London et al. used PAC Bayes analysis to show that for *templated* models, increasing the size of the example actually **decreases** generalization error: in the limit, one could learn an entire templated model from a single example. So, from that perspective, using n as the measure of structure complexity makes no sense, since in most applications (including those in this paper) feature templates are used (bi-grams, etc.). So while I believe that their analysis is technically correct, the authors must reconcile their analysis with previous work in order for this paper to make sense.

- n^4 seems like an awfully large term for a generalization bound: following the supplemental, it seems like it stems from the fact that the bounds rely on decomposing loss linearly in Lemma 6, and then due to having to multiple the norms of 2 examples (which could be O(n)) in addition to that, as well as decomposing the regularization term linearly (O(n/alpha)). To me, that suggests that this is really not the right approach to take here.

- The aforementioned previous work uses a more subtle measure of graph complexity in their bounds, based on concentration inequalities, that measures the maximum dependence of one variable on the others in the graph. It seems like a better approach to analysis would be to relate the novel regularization to the resulting complexity of the learned model in terms of a complexity measure like that, where one can assume some sort of templating. Instead of just assuming complexity = n.

+ I hate to say this (because this idea is so trendy), but the tag structure regularization reminds me a lot of dropout: essentially, for each example, you generate new examples by removing edges in the graph. In the case of sequences, this creates disconnected components, but one could imagine more generally just removing edges. So in that sense I think this paper does help shed light on other ideas in the field.

+ All that being said about the theory, the experimental results are very strong, and the idea is simple enough to be easy to experiment with and verify.
Summary: While the analysis seems like it takes the wrong approach, the idea is simple and interesting, and the experiments are strong. I think it would benefit the community to see it.
Author Feedback
Author rebuttal: We thank the reviewers for insightful comments.

**To Reviewer_1:

1) Is the linear chain assumption a limitation of the method?

>> No, our method does not have a limitation on a linear-chain. The intuition of StructReg is that the information from simple structures has better generalization ability than the information from complex structures, thus StructReg prefers a “simpler” model by penalizing complex structure information. The intuition/method is simple and can naturally extend to graphs. Although we haven’t done experiments on complicated graphs yet, we have experiments on moderately complicated structures with latent variables, and the results confirmed the advantages of our method. We will conduct systematic study on complex structures.

>> Note that, another merit of our method is the inference efficiency, which is more important in graphs, where exact inference may be intractable. StructReg may make the inference exact/efficient.

2) How does this approach relate to other work that approximates structure at training time? E.g…

>> [Sutton & McCallum 07] suggested an interesting piecewise training method via edge-based approximation. This is a related work, but it is substantially different. First, their work focuses on efficiency and there is no indication on generalization. Second, the methodology is different (e.g., deterministic vs. randomized, fixed granularity vs. flexible granularity). Finally, our method has better performance -- on the same POS-tagging dataset, their accuracy is 94.4% (Table 1) and our accuracy is over 97.3%.

>> [Wainwright 2006] suggested consistent approximation for both train/test, but there is no any indication on structure regularization. Thanks for the suggestion. We will make a thorough comparison with related work in the revised version.

3) Some suggestions on removing writing errors and adding additional clarifications …

>> Thanks! We will thoroughly proofread the paper to remove bugs and clarify unclear points.

**To Reviewer_3:

4) One problem that I see is that the provided bounds are much looser than the existing analysis of SO prediction models for hinge-loss …

>> Thanks for the comment. Reviewer_41 has a similar concern. First, we argue that it is *unfair* to directly compare our sample-based bound with prior tag-based bounds on hinge loss [Taskar et al. 2003] and PAC Bayes analysis [London et al. 2013] (this paper is mentioned by Reviewer_41). Actually, our bound should be interpreted as O(n^2) if must make a comparison. Our bound is based on *sample* loss, which is NOT normalized by n (see R and R_e). Differently, the bounds in prior papers are based on *tag* loss, which is normalized by n (see “average per-label loss” in Section 6 of [Taskar 2003] and L(h, Z) in Section 2 of [London 2013]). If must make a comparison, we should also use the tag loss (normalizing R and R_e with n), which will cause O(n) reduction on the stability Delta (via updating Eq.12 in supplemental) and another O(n) reduction on generalization risk (i.e., removing n from Eq.20 via updating Eq.21-22). In other words, this normalization leads to O(n) better stability and O(n) simpler relation between stability and generalization. Thus, if must make a comparison based on prior setting, our bound is O(n^4/n^2) = O(n^2).

>> On the other hand, we agree that improving the bound is possible, and we will show that our bound can be easily corrected to O(n) by relaxing an over-conservative assumption. We assumed extremely dense features, bringing O(n^2) bound on multiplying two feature vectors (see Eq.18 in supplemental). Since many tasks (e.g., the tasks in our paper) have sparse features, it is reasonable to assume sparse features instead, as many prior theoretical work did (e.g., see [Niu et al. Hogwild!, NIPS 2011]). With this very simple debugging, our tag-based bound can be improved from O(n^2) to O(n). By considering “alpha”, our final bound is O(n/alpha).

>> We hope this can clarify the concerns. Since we took a very different analysis approach compared with [Taskar 2003, London 2013], in the future we may reconcile our analysis with prior work for further improvement if possible.

5) … simply looking at the bigram labels yields the best performance (46.6/20) … is this a correct reading?

>> Almost correct. For the 46.6/20 case, the mini-samples are a mixture among bigrams and trigrams, with the distribution center on the mean value.

6) I am assuming here, that during testing the same decomposition algorithm is applied.

>> No decomposition during testing. Like prior regularization methods, we only apply StructReg in training.

7) It would be great if the authors could provide some discussions on these two aspect.

>> Yes, Table 1 is based on test data. The hyper-parameters are tuned on development/held-out data. We did not show the tuning due to space limitation. We will include the tuning of hyper-parameters in the supplementary file. We will also release all code for reproducibility.

8) what are the consequence of the random decomposition approach for models with clique size larger than 2.

>> Please refer to 1) from Reviewer_1 for our reply. Thanks.

**To Reviewer_41:

9) …the bound is still far looser than other structured prediction generalization bounds ... London et al. used PAC Bayes analysis to show that …

>> Reviewer_3 has a similar comment. Please refer to 4) for our reply. Thanks.

10) the tag structure regularization reminds me a lot of dropout …

>> Thanks for the insightful comment. Yes, edge-removing is an interesting extension. Also, due to space limitation, some extra merits of StructReg were not shown – one is the parallelized training over a *single* sample (parallelization over mini-samples). It can speed up the SGD training 6 times based on a CPU with 8 cores, and without accuracy loss. We will add a discussion in the revised paper.